# Unsupervised Meta-Learning through Latent-Space Interpolation in Generative Models

**Siavash Khodadadeh**[1] *  **Sharare Zehtabian**[1]*  **Saeed Vahidian**[2]

**Weijia Wang**[2]  **Bill Lin**[2]  **Ladislau Bölöni**[1]

[1] Dept. of Computer Science
University of Central Florida

[2] Dept. of Electrical & Computer Engineering
University of California San Diego

## Abstract

Several recently proposed unsupervised meta-learning approaches rely on synthetic meta-tasks created using techniques such as random selection, clustering and/or augmentation. In this work, we describe a novel approach that generates meta-tasks using generative models. The proposed family of algorithms generate pairs of in-class and out-of-class samples from the latent space in a principled way, allowing us to create synthetic classes forming the training and validation data of a meta-task. We find that the proposed approach, LAtent Space Interpolation Unsupervised Meta-learning (LASIUM), outperforms or is competitive with current unsupervised learning baselines on few-shot classification tasks on the most widely used benchmark datasets.

## 1 Introduction

Few-shot meta-learning algorithms for neural networks such as Mishra et al. (2018); Finn et al. (2017); Snell et al. (2017) prepare networks to quickly adapt to unseen tasks. This is done in a meta-training phase that typically involves a large number of supervised learning tasks. Generating supervised tasks depends on large, labeled datasets and hand-specified task distribution. Very recently, several approaches had been proposed that perform the meta-training by generating synthetic training tasks from an underlined dataset. This requires us to generate samples with specific pairwise information: in-class pairs of samples that are with high likelihood in the same class, and out-of-class pairs that are with high likelihood not in the same class. For instance, UMTRA by Khodadadeh et al. (2019) and AAL by Antoniou & Storkey (2019) achieve this through random selection from a domain with many classes for out-of-class pairs and by augmentation for in-class pairs. Hsu et al. (2019) propose CACTUs that creates synthetic labels through unsupervised clustering of the domain.

In this paper, we rely on recent advances in the field of generative models, such as the variants of generative adversarial networks (GANs) and variational autoencoders (VAEs), to generate the in-class and out-of-class pairs of meta-training data. The fundamental idea of our approach is that in-class pairs are close while out-of-class pairs are far away in the latent space representation of the generative model. Thus, we can generate in-class pairs by interpolating between two out-of-class samples in the latent space and choosing interpolation ratios that put the new sample close to one of the objects. From this latent sample, the generative model creates the new in-class object. Our approach requires little domain-specific tweaking, and the necessary tweaks are human-comprehensible. For instance, we need to choose thresholds for latent space distance that ensure that classes are in different domains, as well as interpolation ratio thresholds that ensure that the sample is in the same class as the nearest edge. Another advantage of the approach is that we can utilize off-the-shelf, pre-trained generative models. The main contributions of this paper can be summarized as follows:

---

*Equal contribution. Correspondence to {siavash.khodadadeh, sharare.zehtabian}@knights.ucf.edu

- We describe an algorithm, LAtent Space Interpolation Unsupervised Meta-learning (LA-SIUM), that creates training data for a downstream meta-learning algorithm starting from an unlabeled dataset by leveraging interpolation in the latent space of a generative model.

- We show that on the most widely used few-shot learning datasets, LASIUM outperforms or performs competitively with other unsupervised meta-learning algorithms, significantly outperforms transfer learning in all cases, and in a number of cases approaches the performance of supervised meta-learning algorithms.

## 2 RELATED WORK

Meta-learning or "learning to learn" in the field of neural networks is an umbrella term that covers a variety of techniques that involve training a neural network over the course of a meta-training phase, such that when presented with the target task, the network is able to learn it much more efficiently than a randomly initialized network would. Such techniques had been proposed since the 1980s (Schmidhuber (1987); Bengio et al. (1990); Naik & Mammone (1992); Thrun & Pratt (1998)). In recent years, meta-learning has gained a resurgence, through approaches that either "learn to optimize" (Finn et al. (2017); Ravi & Larochelle (2016); Mishra et al. (2017); Nichol et al. (2018); Rusu et al. (2019); Rajeswaran et al. (2019)) or learn embedding functions in a non-parametric setting (Snell et al. (2017); Vinyals et al. (2016); Ren et al. (2018); Liu et al. (2019)). Hybrids between these two approaches had also been proposed (Triantafillou et al. (2020); Wang et al. (2019)).

Most approaches use labeled data during the meta-learning phase. While in some domains there is an abundance of labeled datasets, in many domains such labeled data is difficult to acquire. Unsupervised meta-learning approaches aim to learn from an unsupervised dataset from a domain similar from that of the target task. Typically these approaches generate synthetic few-shot learning tasks for the meta-learning phase through a variety of techniques. CACTUs (Hsu et al. (2019)) uses a progressive clustering method. UMTRA (Khodadadeh et al. (2019)) utilizes the statistical diversity properties and domain-specific augmentations to generate synthetic training and validation data. AAL (Antoniou & Storkey (2019)) uses augmentation of the unlabeled training set to generate the validation data. The accuracy of these approaches was shown to be comparable with but lower than supervised meta-learning approaches, though with the advantage of requiring orders of magnitude less labeled training data. A common weakness of these approaches is that the techniques used to generate the synthetic tasks (clustering, augmentation, random sampling) are highly domain dependent.

Our proposed approach, LASIUM, takes advantage of generative models trained on the specific domain to create the in-class and out-of-class pairs of meta-training data. By creating new training data through interpolation between training samples, LASIUM-OC has similarities with mixup Zhang et al. (2017). The most successful neural-network based generative models in recent years are variational autoencoders (VAE) (Diederik & Welling (2014)) and generative adversarial networks (GANs) (Goodfellow et al. (2014)). The implementation variants of the LASIUM algorithm described in this paper rely on the original VAE model and on two specific variations of the GAN concept, respectively. MSGAN (aka Miss-GAN) (Mao et al. (2019)) aims to solve the missing mode problem of conditional GANs through a regularization term that maximizes the distance between the generated images with respect to the distance between their corresponding input latent codes. Progressive GAfNs (Karras et al. (2018)) are growing both the generator and discriminator progressively, and approach resembling the layer-wise training of autoencoders.

## 3 LATENT SPACE INTERPOLATION UNSUPERVISED META-LEARNING

**Preliminaries**: We define an $N$-way, $K^{(tr)}$-shot supervised classification task, $\mathcal{T}$, as a set $\mathcal{D}_{\mathcal{T}}^{(tr)}$ composed of $i \in \{1, \dots, N \times K^{(tr)}\}$ data points $(x_i, y_i)$ such that there are exactly $K^{(tr)}$ samples for each categorical label $y_i \in \{1, \dots, N\}$. During meta-learning, an additional set ,$\mathcal{D}_{\mathcal{T}}^{(val)}$, is attached to each task that contains another $N \times K^{(val)}$ data points separate from the ones in $\mathcal{D}_{\mathcal{T}}^{(tr)}$. We have exactly $K^{(val)}$ samples for each class in $\mathcal{D}_{\mathcal{T}}^{(val)}$ as well.

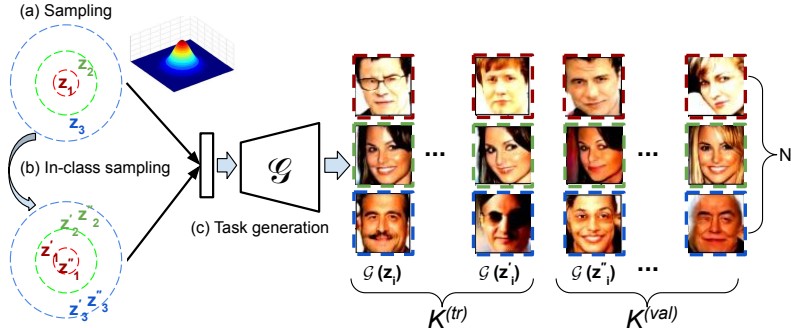

Figure 1: 3-way, $K^{(tr)}$-shot task generation with $K^{(val)}$ images for validation by a pre-trained GAN generator $\mathcal{G}$. **a)** Sample 3 random vectors. **b)** Generate new vectors by one of the proposed in-class sampling strategies. **c)** Generate images from all of the latent vectors and put them into train and validation set to construct a task. The images in this figure have been generated by our algorithm. The colored edge of each image indicates that it was generated from its corresponding latent vector.

It is straightforward to package $N$-way, $K^{(tr)}$-shot tasks with $\mathcal{D}_{\mathcal{T}}^{(tr)}$ and $\mathcal{D}_{\mathcal{T}}^{(val)}$ from a labeled dataset. However, in unsupervised meta-learning setting, a key challenge is how to automatically construct tasks from the unlabeled dataset $\mathcal{U} = \{x_i\}$.

## 3.1 GENERATING META-TASKS USING GENERATIVE MODELS

We have seen that in order to generate the training data for the meta-learning phase, we need to generate $N$-way training tasks with $K^{(tr)}$ training and $K^{(val)}$ validation samples. The label associated with the classes in these tasks is not relevant, as it will be discarded after the meta-learning phase. Our objective is simply to generate samples of the type $x_{i,j}$ with $i \in \{1 \ldots N\}$ and $j \in \{1 \ldots K^{(tr)} + K^{(val)}\}$ with the following properties: (a) all the samples $x_{i,j}$ are different (b) any two samples with the same $i$ index are in-class samples and (c) any two samples with different $i$ index are out-of-class samples. In the absence of human provided labels, the class structure of the domain is defined only implicitly by the sample selection procedure. Previous approaches to unsupervised meta-learning chose samples directly from the training data $x_{i,j} \in \mathcal{U}$, or created new samples through augmentation. For instance, we can define the class structure of the domain by assuming that certain types of augmentations keep the samples in-class with the original sample. One challenge of such approaches is that the choice of the augmentation is domain dependent, and the augmentation itself can be a complex mathematical operation.

In this paper, we approach the sample selection problem differently. Instead of sampling $x_{i,j}$ from $\mathcal{U}$, we use the unsupervised dataset to train a generative model $p(x)$. Generative models represent the full probability distribution of a model, and allow us to sample new instances from the distribution. For many models, this sampling process can be computationally expensive iterative process. Many successful neural network based generative models use the reparametrization trick for the training and sampling which concentrate the random component of the model in a latent representation $z$. By choosing the latent representation $z$ from a simple (uniform or normal) distribution, we can obtain a sample from the complex distribution $p(x)$ by passing $z$ through a deterministic generator $\mathcal{G}(\mathcal{Z}) \rightarrow \mathcal{X}$. Two of the most popular generative models, variational autoencoders (VAEs) and generative adversarial networks (GANs) follow this model.

The idea of the LASIUM algorithm is that given a generator $\mathcal{G}(.)$, nearby latent space values $z_1$ and $z_2$ map to in-class samples $x_1$ and $x_2$ that belong to the same class. Conversely, $z_1$ and $z_2$ values that are far away from each other, map to out of class samples that belong to different classes. Naturally, we still need to define what we mean by "near" and "far" in the latent space and how to choose the corresponding $z$ values. However, this is a significantly simpler task than, for instance, defining the set of complex augmentations that might retain class membership.

**Training a generative model** Our method for generating meta-tasks is agnostic to the choice of training algorithm for the generative model and can use either a VAE or a GAN with the only

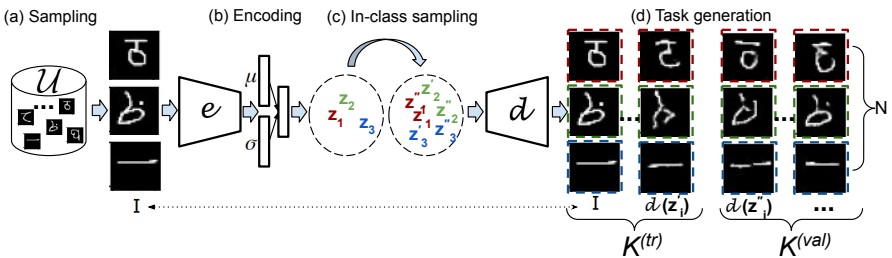

Figure 2: 3-way, $K^{(tr)}$-shot task generation by VAE on Omniglot dataset with $K^{(val)}$ images for validation set of each task. **a)** Sample 3 images from dataset. **b)** Encode the images into latent space and check if they are distanced. **c)** Use proposed in-class sampling techniques to generate new latent vectors. **d)** Generate images from the latent vectors and put them alongside with sampled images from step **a** into train and validation set to construct a task.

---

**Algorithm 1:** LASIUM for unsupervised meta-learning task generation

---

**require :** Unlabeled dataset $\mathcal{U} = \{x_i\}$, Pre-trained generator $\mathcal{G}$
**require :** $N$: class-count, $N_{MB}$: meta-batch size, $\epsilon$: minimum threshold, $\sigma$ or $\alpha$: sampling hyperparameters
**require :** Sampling Strategy $\mathcal{S}$: could be $N(\sigma^2)$, $RO(\alpha)$, or $OC(\alpha)$
**require :** $K^{(tr)}$, $K^{(val)}$: number of samples for train and validation during meta-learning

1   $B = \{\}$ ; `// meta-batch of tasks`
2   **for** i in $1, \ldots, N_{MB}$ **do**
3     Sample $N$ class-vectors in latent space of $\mathcal{G}$ and add them to task-vectors until they are at least $\epsilon$ units away from each other in euclidean space
4     **for** $\omega$ in $1, \ldots, K^{(tr)} + K^{(val)} - 1$ **do**
5       Generate new-vectors = $\mathcal{S}$(class-vectors, $\omega$) and add them to task-vectors
6     **end**
7     Generate $N \times (K^{(tr)} + K^{(val)})$ images by feeding task-vectors to generator $\mathcal{G}$
8     Construct task $\mathcal{T}_i$ by putting the first $N \times K^{(tr)}$ images in task train set and the last $N \times K^{(val)}$ images in task validation set
9     $B \leftarrow B \cup \mathcal{T}_i$
10 **end**
11 **return** B

---

constraint of having appropriately structured latent space. In our VAE experiments, we used a network trained with the standard VAE training algorithm (Diederik & Welling (2014)). For the experiments with GANs we used two different methods mode seeking GANs (MSGAN) (Mao et al. (2019)) and progressive growing of GANs (ProGAN) (Karras et al. (2018)). MSGAN is trained for Omniglot and ProGAN is trained for CelebA.

Algorithm 1 describes the steps of our method. We will delve into each step in the following parts of this section.

**Sampling out of class instances from the latent space representation:** Our sampling techniques differ slightly whether we are using a GAN or VAE. For GAN, we use rejection sampling to find $N$ latent space vectors that are at a pairwise distance of at least threshold $\epsilon$ - see Figure 1(a). When using a VAE, we also have an encoder network that allows us to map from the domain to the latent space. Taking advantage of this, we can additionally sample data points from our unlabeled dataset $\mathcal{U}$ and embed them into a latent space. If the latent space representation of these $N$ images are too close to each other, we re-sample, otherwise we can use the $N$ images and their representations and continue the following steps exactly the same as GANs - see Figure 2(a) and (b). We will refer to the vectors selected here as underline{anchor vectors}.

**Generating in-class latent space vectors** Next, having $N$ sampled anchor vectors $\{z_1, \ldots, z_N\}$ from the latent space representation, we aim to generate $N$ new vectors $\{z'_1, \ldots, z'_N\}$ from the latent

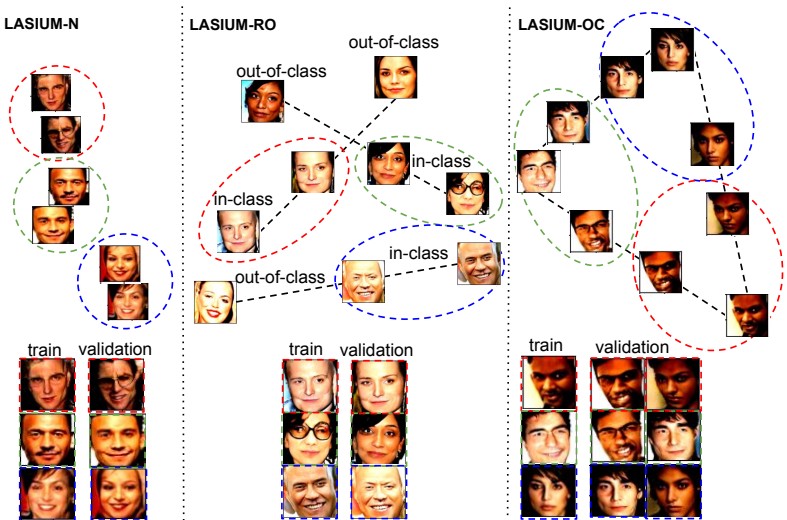

Figure 3: Latent space representation visualization of proposed strategies for generating in-class candidates. **Left**: LASIUM-N, adding random noise to the sample vector. **Middle**: LASIUM-RO, interpolate with random out-of-class samples. **Right**: LASIUM-OC, interpolate with other classes' samples.

space representation such that the generated image $\mathcal{G}(z_i)$ belongs to the same class as the one of $\mathcal{G}(z_i')$ for $i \in 1, \ldots, N$. This process needs to be repeated for $K^{(tr)} + K^{(val)} - 1$ times.

The sampling strategy takes as input the sampled vectors and a number $\omega \in \{1 \ldots K^{(tr)} + K^{(val)} - 1\}$ and returns $N$ new vectors such that $z_i$ and $z_i'$ are an in-class pair for $i \in \{1 \ldots N\}$. This ensures that no two $z_i'$ belong to the same class and creates $N$ groups of $(K^{(tr)} + K^{(val)})$ vectors in our latent space. We feed these vectors to our generator to get $N$ groups of $(K^{(tr)} + K^{(val)})$ images. From each group we pick the first $K^{(tr)}$ for $\mathcal{D}_{\mathcal{T}}^{(tr)}$ and the last $K^{(val)}$ for $\mathcal{D}_{\mathcal{T}}^{(val)}$.

What remains is to define the strategy to sample the individual in-class vectors. We propose three different sampling strategies, all of which can be seen as variations of the idea of latent space interpolation sampling. This motivates the name of the algorithm LAtent Space Interpolation Unsupervised Meta-learning (LASIUM).

**LASIUM-N (adding Noise)**: This technique generates in-class samples by adding Gaussian noise to the anchor vector $z_i' = z_i + \epsilon$ where $\epsilon \sim \mathcal{N}(0, \sigma^2)$ (see Figure 3-Left). In the context of LASIUM, we can see this as an interpolation between the anchor vector and a noise vector, with the interpolation factor determined by $\sigma$. For the impact of different choices of $\sigma$ see the ablation study in the supplemental material.

**LASIUM-RO (with Random Out-of-class samples)** To generate a new in-class sample to anchor vector $z_i$ we first find a random out-of-class sample $v_i$, and choose an interpolated version closer to the anchor: $z_i' = z_i + \alpha \times (v_i - z_i)$ (see Figure 3-Middle). Here, $\alpha$ is a hyperparameter, which can be tuned to define the size of the class. As we are in a comparatively high-dimensional latent space (in our case, 512 dimensions), we need relatively large values of $\alpha$, such as $\alpha = 0.4$ to define classes of reasonable size. This model effectively allows us to define complex augmentations (such as a person seen without glasses, or in a changed lighting) with only one scalar hyperparameter to tune. By interpolating towards another sample we ensure that we are staying on the manifold that defines the dataset (in the case of Figure 3, this being human faces).

**LASIUM-OC (with Other Classes' samples)** This technique is similar to LASIUM-RO, but instead of using a randomly generated out-of-class vector, we are interpolating towards vectors already chosen from the other classes in the same task (see Figure 3-Right). This limits the selection of the samples to be confined to subspace of the convex hull containing the initial anchor points. The intuition behind this approach is that choosing the samples this way focuses the attention of the meta-learner towards the hard to distinguish samples that are between the classes in the few shot learning class (eg. they share certain attributes).

Finally, failure cases exist for these sampling strategies. For example, if $z_1$, $z_2$, and $z_3$ lie on a line, we end up generating poor out of class samples that belong to other classes in our task. However, as we are in high dimensional latent space(e.g. 512), the likelihood for three points being approximately colliniar is very low (as the volume fraction in which the middle point would have to be shrinks exponentially with the dimensionality). Note that meta-learning algorithms are not sensitive to a small fraction of bad training examples in the meta-training phase (e.g. see Hsu et al. (2019); Khodadadeh et al. (2019)).

# 4 EXPERIMENTAL VALIDATION

We tested the proposed algorithms on three few-shot learning benchmarks: (a) the 5-way Omniglot (Lake et al. (2011)), a benchmark for few-shot handwritten character recognition, (b) the 5-way CelebA few-shot identity recognition, and (c) the CelebA attributes dataset (Liu et al. (2015)) proposed as a few-shot learning benchmark by (Hsu et al. (2019)). The latter benchmark comprises binary classification (2-way) tasks in which each task is defined by selecting 3 different attributes and 3 boolean values corresponding to each attribute. Every image in a certain task-specific class has the same attributes with each other while does not share any of these attributes with images in the other class.

We partition each dataset into meta-training, meta-validation, and meta-testing splits between classes. To evaluate our method, we use the classes in the test set to generate 1000 tasks as described in section 3. We set $K^{(val)}$ to be 15. Note that unlike meta-training, where we use synthetic generated tasks, for evaluation we use 1000 tasks from real unseen labeled data. Furthermore, we fix the random seed to make sure we compare on the exact same tasks with other baselines. We average the accuracy on all tasks and report a $95\%$ confidence interval. To ensure that comparisons are fair, we use the same random seed in the whole task generation process. For the Omniglot dataset, we report the results for $K^{(tr)} \in \{1, 5\}$, and $K^{(val)} = 15$. For CelebA identity recognition, we report our results for $K^{(tr)} \in \{1, 5, 15\}$ and $K^{(val)} = 15$. For CelebA attributes, we follow the $K^{(tr)} = 5$ and $K^{(val)} = 5$ tasks as proposed by Hsu et al. (2019). More ablation studies over the hyperparameters and result visualizations are provided in the supplemental material. Since excessive tuning of hyperparameters can lead to the overestimation of the performance of a model (Oliver et al. (2018)), we keep the hyperparameters of the unsupervised meta-learning as constant as possible (including the MAML, and ProtoNets model architectures) in all experiments. Our model architecture consists of four stacked convolutional blocks. Each block comprises 64 filters that carry out $3 \times 3$ convolutions, followed by batch normalization, a ReLU non-linearity, and $2 \times 2$ max-pooling. For the MAML experiments, classification is performed by a fully connected layer, whereas for the ProtoNets model we compute distances based on the feature vectors produced by the last convolution module without any dense layers. The input size to our model is $84 \times 84 \times 3$ for CelebA and $28 \times 28 \times 1$ for Omniglot. The detail of the neural networks architectures for each experiment is described in the supplemental material.

## 4.1 BASELINES

As baseline algorithms for our approach we follow the practice of recent papers in the unsupervised meta-learning literature. The simplest baseline is to train the same network architecture from scratch with $N \times K^{(tr)}$ images. More advanced baselines can be obtained by learning an unsupervised embedding on $\mathcal{U}$ and use it for downstream task training. We used the ACAI (Berthelot et al. (2019)), BiGAN (Donahue et al. (2017); Dumoulin et al. (2017)), and DeepCluster (Caron et al. (2018)) as representative of the unsupervised learning literature. On top of these embeddings, we report accuracy for $K_{nn}$-nearest neighbors, linear classifier, multi layer perceptron (MLP) with dropout, and cluster matching.

The direct competition for our approach are the current state-of-the-art algorithms in unsupervised meta-learning. We compare our results with CACTUs-MAML, CACTUs-ProtoNets (Hsu et al. (2019)) and UMTRA Khodadadeh et al. (2019). Finally, it is useful to compare our approach with algorithms that require supervised data. We include results for supervised standard transfer learning from VGG19 pre-trained on ImageNet (Simonyan & Zisserman (2015)) and two supervised meta-learning algorithms, MAML (Finn et al. (2017)), and ProtoNets (Snell et al. (2017)).

Table 1: Accuracy results on the Omniglot dataset averaged over 1000, 5-way, $K^{(tr)}$-shot downstream tasks with $K^{(val)} = 15$ for each task. $\pm$ indicates the 95% confidence interval. The top three unsupervised results are reported in **bold**. The baseline results are from Hsu et al. (2019) section 4.1.

| Algorithm | Feature Extractor | $K^{(tr)} = 1$ | $K^{(tr)} = 5$ |
|---|---|---|---|
| Training from scratch | $N/A$ | $51.64 \pm 0.65$ | $71.44 \pm 0.53$ |
| K-nearest neighbors | ACAI | $57.46 \pm 1.35$ | $81.16 \pm 0.57$ |
| Linear Classifier | ACAI | $61.08 \pm 1.32$ | $81.82 \pm 0.58$ |
| MLP with dropout | ACAI | $51.95 \pm 0.82$ | $77.20 \pm 0.65$ |
| Cluster matching | ACAI | $54.94 \pm 0.85$ | $71.09 \pm 0.77$ |
| K-nearest neighbors | BiGAN | $49.55 \pm 1.27$ | $68.06 \pm 0.71$ |
| Linear Classifier | BiGAN | $48.28 \pm 1.25$ | $68.72 \pm 0.66$ |
| MLP with dropout | BiGAN | $40.54 \pm 0.79$ | $62.56 \pm 0.79$ |
| Cluster matching | BiGAN | $43.96 \pm 0.80$ | $58.62 \pm 0.78$ |
| CACTUs-MAML | BiGAN | $58.18 \pm 0.81$ | $78.66 \pm 0.65$ |
| CACTUs-MAML | ACAI | $68.84 \pm 0.80$ | $87.78 \pm 0.50$ |
| UMTRA-MAML | $N/A$ | $\mathbf{81.91 \pm 0.58}$ | $\mathbf{94.58 \pm 0.25}$ |
| LASIUM-RO-GAN-MAML | $N/A$ | $\mathbf{83.26 \pm 0.55}$ | $\mathbf{95.29 \pm 0.22}$ |
| LASIUM-N-VAE-MAML | $N/A$ | $76.11 \pm 0.64$ | $\mathbf{94.42 \pm 0.26}$ |
| CACTUs-ProtoNets | BiGAN | $54.74 \pm 0.82$ | $71.69 \pm 0.73$ |
| CACTUs-ProtoNets | ACAI | $68.12 \pm 0.84$ | $83.58 \pm 0.61$ |
| LASIUM-RO-GAN-ProtoNets | $N/A$ | $\mathbf{80.15 \pm 0.64}$ | $91.10 \pm 0.35$ |
| LASIUM-OC-VAE-ProtoNets | $N/A$ | $73.22 \pm 0.73$ | $85.05 \pm 0.46$ |
| Transfer Learning (VGG-19) | $N/A$ | $54.49 \pm 0.90$ | $89.57 \pm 0.44$ |
| Supervised MAML | $N/A$ | $94.46 \pm 0.35$ | $98.83 \pm 0.12$ |
| Supervised ProtoNets | $N/A$ | $98.35 \pm 0.22$ | $99.58 \pm 0.09$ |

## 4.2 RESULTS ON OMNIGLOT

Table 1 shows the results on the Omniglot dataset. We find that the LASIUM-RO-GAN-MAML configuration outperforms all the unsupervised approaches, including the meta-learning based ones like CACTUs and UMTRA. Beyond the increase in performance, we must note that the (one-dimensional) search for the intra-class shift was much cheaper than fine-tuning the currently popular augmentation strategies. We also find that on this benchmark, LASIUM outperforms transfer learning using the much larger VGG-19 network.

As expected even the best LASIUM result is worse than the supervised meta-learning models. However, we need to consider that the unsupervised meta-learning approaches use several orders of magnitude less labels. For instance, the 95.29% accuracy of LASIUM-RO-GAN-MAML was obtained with only 25 labels, while the supervised approaches used 25,000.

## 4.3 RESULTS ON CELEBA

Table 2 shows our results on the CelebA identity recognition tasks where the objective is to recognize $N$ different people given $K^{(tr)}$ images for each. We find that on this benchmark as well, the LASIUM-RO-GAN-MAML configuration performs better than other unsupervised meta-learning models as well as transfer learning with VGG-19 - it only falls slightly behind LASIUM-RO-GAN-ProtoNets on the one-shot case. As we have discussed in the case of Omniglot results, the performance remains lower then the supervised meta-learning approaches which use several orders of magnitude more labeled data.

Finally, Table 3 shows our results for CelebA attributes benchmark introduced in (Hsu et al. (2019)). A peculiarity of this dataset is that the way in which classes are defined based on the attributes, the classes are unbalanced in the dataset, making the job of synthetic task selection more difficult. We find that LASIUM-N-GAN-MAML obtains a performance of $\mathbf{75.07 \pm 1.08}$, within the confidence interval of the second best, CACTUs MAML with BiGAN $\mathbf{74.98 \pm 1.02}$. In this benchmark, transfer learning with the VGG-19 network performed better than all unsupervised meta-learning approaches,

Table 2: Accuracy results of unsupervised learning on CelebA for different unsupervised methods. The results are averaged over 1000, 5-way, $K^{(tr)}$-shot downstream tasks with $K^{(val)} = 15$ for each task. $\pm$ indicates the 95% confidence interval. The top three unsupervised results are reported in **bold**.

| Algorithm | $K^{(tr)} = 1$ | $K^{(tr)} = 5$ | $K^{(tr)} = 15$ |
|---|---|---|---|
| Training from scratch | $34.69 \pm 0.50$ | $56.50 \pm 0.55$ | $70.56 \pm 0.49$ |
| CACTUs | $41.42 \pm 0.64$ | $\mathbf{62.71 \pm 0.57}$ | $\mathbf{74.18 \pm 0.68}$ |
| UMTRA | $39.30 \pm 0.59$ | $60.44 \pm 0.56$ | $\mathbf{72.41 \pm 0.48}$ |
| LASIUM-RO-GAN-MAML | $\mathbf{43.88 \pm 0.57}$ | $\mathbf{66.98 \pm 0.53}$ | $\mathbf{78.13 \pm 0.44}$ |
| LASIUM-RO-VAE-MAML | $41.25 \pm 0.57$ | $58.22 \pm 0.54$ | $71.05 \pm 0.49$ |
| LASIUM-RO-GAN-ProtoNets | $\mathbf{44.39 \pm 0.61}$ | $60.83 \pm 0.58$ | $66.66 \pm 0.53$ |
| LASIUM-RO-VAE-ProtoNets | $\mathbf{43.22 \pm 0.58}$ | $\mathbf{61.12 \pm 0.54}$ | $68.51 \pm 0.51$ |
| Transfer Learning (VGG-19) | $33.28 \pm 0.57$ | $58.74 \pm 0.62$ | $74.04 \pm 0.49$ |
| Supervised MAML | $85.46 \pm 0.55$ | $94.98 \pm 0.25$ | $96.18 \pm 0.19$ |
| Supervised ProtoNets | $84.17 \pm 0.61$ | $90.84 \pm 0.38$ | $90.85 \pm 0.36$ |

Table 3: Results on CelebA attributes benchmark 2-way, 5-shot tasks with $K^{(val)} = 5$. The results are averaged over 1000 downstream tasks and $\pm$ indicates 95% confidence interval. The top three unsupervised results are reported in **bold**. The baseline results are from Hsu et al. (2019) section 4.1.

| Algorithm | Feature Extractor | Accuracy |
|---|---|---|
| Training from scratch | N/A | $63.19 \pm 1.06$ |
| K-nearest neighbors | BiGAN | $56.15 \pm 0.89$ |
| Linear Classifier | BiGAN | $58.44 \pm 0.90$ |
| MLP with dropout | BiGAN | $56.26 \pm 0.94$ |
| Cluster matching | BiGAN | $56.20 \pm 1.00$ |
| K-nearest neighbors | DeepCluster | $61.47 \pm 0.99$ |
| Linear Classifier | DeepCluster | $59.57 \pm 0.98$ |
| MLP with dropout | DeepCluster | $60.65 \pm 0.98$ |
| Cluster matching | DeepCluster | $51.51 \pm 0.89$ |
| CACTUs MAML | BiGAN | $\mathbf{74.98 \pm 1.02}$ |
| CACTUs MAML | DeepCluster | $\mathbf{73.79 \pm 1.01}$ |
| LASIUM-N-GAN-MAML | N/A | $\mathbf{75.07 \pm 1.08}$ |
| CACTUs ProtoNets | BiGAN | $65.58 \pm 1.04$ |
| CACTUs ProtoNets | DeepCluster | $74.15 \pm 1.02$ |
| LASIUM-N-GAN-ProtoNets | N/A | $73.41 \pm 1.10$ |
| Transfer Learning (VGG-19) | N/A | $79.76 \pm 1.03$ |
| Supervised MAML | N/A | $87.10 \pm 0.85$ |
| Supervised ProtoNets | N/A | $85.13 \pm 0.92$ |

possibly due to existing representations of the discriminating attributes in that much more complex network.

## 4.4 RESULTS ON MINI-IMAGENET

Table 4 shows the comparison results on the Mini-ImageNet benchmark. Mini-ImageNet has a large sample fidelity among the datasets considered in this paper, and it is hard for generative models to capture this diversity given a relatively small dataset such as Mini-ImageNet train set. Thus, we used as the GAN a pre-trained BigBiGAN[1] trained on the whole Imagenet dataset with no supervision. Examples of meta-training tasks constructed by LASIUM-N with $\sigma^2 = 1.0$ are shown in Figure 4. We notice that LASIUM-N-GAN-MAML outperforms all other unsupervised learning algorithms for K=1, 5 and 20. For K=50, it is in the second place behind CACTUs MAML with DeepCluster, the accuracy difference being within the margin of error.

---

[1]https://tfhub.dev/deepmind/bigbigan-resnet50/1

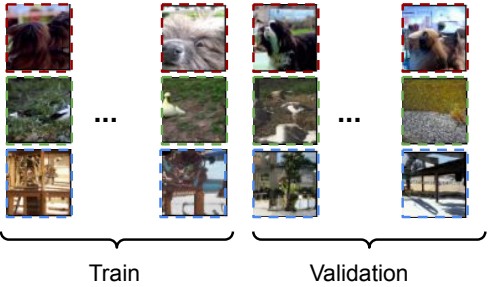

Figure 4: Meta-training tasks for Mini-ImageNet constructed by LASIUM-N with $\sigma^2 = 1.0$

Table 4: Results on Mini-Imagenet benchmark for 5-way, $K^{(tr)}$-shot tasks with $K^{(val)} = 15$. The results are averaged over 1000 downstream tasks and $\pm$ indicates 95% confidence interval. The top three unsupervised results are reported in **bold**. The baselines are from Hsu et al. (2019) section 4.1. Note that the BigBiGAN is trained on the unlabeled Imagenet dataset which is larger than Mini-ImageNet training set that used by Hsu et al. (2019) for training.

| Algorithm | Embedding | $K^{(tr)} = 1$ | $K^{(tr)} = 5$ | $K^{(tr)} = 20$ | $K^{(tr)} = 50$ |
|---|---|---|---|---|---|
| Training from scratch | N/A | $27.59 \pm 0.59$ | $38.48 \pm 0.66$ | $51.53 \pm 0.72$ | $59.63 \pm 0.74$ |
| K-nearest neighbors | BiGAN | $25.56 \pm 1.08$ | $31.10 \pm 0.63$ | $37.31 \pm 0.40$ | $43.60 \pm 0.37$ |
| Linear Classifier | BiGAN | $27.08 \pm 1.24$ | $33.91 \pm 0.64$ | $44.00 \pm 0.45$ | $50.41 \pm 0.37$ |
| MLP with dropout | BiGAN | $22.91 \pm 0.54$ | $29.06 \pm 0.63$ | $40.06 \pm 0.72$ | $48.36 \pm 0.71$ |
| Cluster matching | BiGAN | $24.63 \pm 0.56$ | $29.49 \pm 0.58$ | $33.89 \pm 0.63$ | $36.13 \pm 0.64$ |
| K-nearest neighbors | DeepCluster | $28.90 \pm 1.25$ | $42.25 \pm 0.67$ | $56.44 \pm 0.43$ | $63.90 \pm 0.38$ |
| Linear Classifier | DeepCluster | $29.44 \pm 1.22$ | $39.79 \pm 0.64$ | $56.19 \pm 0.43$ | $65.28 \pm 0.34$ |
| MLP with dropout | DeepCluster | $29.03 \pm 0.61$ | $39.67 \pm 0.69$ | $52.71 \pm 0.62$ | $60.95 \pm 0.63$ |
| Cluster matching | DeepCluster | $22.20 \pm 0.50$ | $23.50 \pm 0.52$ | $24.97 \pm 0.54$ | $26.87 \pm 0.55$ |
| CACTUs MAML | BiGAN | $36.24 \pm 0.74$ | $51.28 \pm 0.68$ | $61.33 \pm 0.67$ | $66.91 \pm 0.68$ |
| CACTUs MAML | DeepCluster | $39.90 \pm 0.74$ | $\mathbf{53.97 \pm 0.70}$ | $\mathbf{63.84 \pm 0.70}$ | $\mathbf{69.64 \pm 0.63}$ |
| UMTRA MAML | N/A | $\mathbf{39.93}$ | $50.73$ | $61.11$ | $\mathbf{67.15}$ |
| LASIUM-N-GAN-MAML | N/A | $\mathbf{40.19 \pm 0.58}$ | $\mathbf{54.56 \pm 0.55}$ | $\mathbf{65.17 \pm 0.49}$ | $\mathbf{69.13 \pm 0.49}$ |
| CACTUs ProtoNets | BiGAN | $36.62 \pm 0.70$ | $50.16 \pm 0.73$ | $59.56 \pm 0.68$ | $63.27 \pm 0.67$ |
| CACTUs ProtoNets | DeepCluster | $39.18 \pm 0.71$ | $\mathbf{53.36 \pm 0.70}$ | $\mathbf{61.54 \pm 0.68}$ | $63.55 \pm 0.64$ |
| LASIUM-N-GAN-ProtoNets | N/A | $\mathbf{40.05 \pm 0.60}$ | $52.53 \pm 0.51$ | $59.45 \pm 0.48$ | $61.43 \pm 0.45$ |
| Transfer Learning | N/A | $44.06 \pm 0.66$ | $70.11 \pm 0.67$ | $86.12 \pm 0.36$ | $92.67 \pm 0.22$ |
| Supervised MAML | N/A | $46.81 \pm 0.77$ | $62.13 \pm 0.72$ | $71.03 \pm 0.69$ | $75.54 \pm 0.62$ |
| Supervised ProtoNets | N/A | $46.56 \pm 0.76$ | $62.29 \pm 0.71$ | $70.05 \pm 0.65$ | $72.04 \pm 0.60$ |

## 5 DISCUSSION

We described LASIUM, an unsupervised meta-learning algorithm for few-shot classification. The algorithm creates synthetic meta-tasks using interpolation in the latent space of a generative model with the general idea that points that are close in the latent space will likely generate in-class samples, while points far in the latent space will likely generate out-of-class samples. In this, LASIUM differs from techniques such as UMTRA and AAL that generate in-class samples through pixel-space augmentation and techniques such as CACTUs which uses clustering to separate unsupervised training data into in-class and out-of-class samples. We found that LASIUM outperforms or comes within the margin of error to state-of-the-art unsupervised algorithms on Omniglot, Mini-ImageNet, CelebA attributes learning benchmark and the CelebA identity recognition benchmarks.

## 6 ACKNOWLEDGEMENTS

This work had been in part supported by the National Science Foundation under Grant Number IIS-1409823.

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

## 7 HYPERPARAMETERS AND ABLATION STUDIES

In this section, we report the hyperparameters of LASIUM-MAML in Table 5 and LASIUM-ProtoNets in Table 6 for Omniglot, CelebA, CelebA attributes and Mini-ImageNet datasets. Our source code is also available on Github [2].

Table 5: LASIUM-MAML hyperparameters summary

| Hyperparameter | Omniglot | CelebA | CelebA attributes | Mini-ImageNet |
|---|---|---|---|---|
| Number of classes | 5 | 5 | 2 | 5 |
| Input size | $28 \times 28 \times 1$ | $84 \times 84 \times 3$ | $84 \times 84 \times 3$ | $84 \times 84 \times 3$ |
| Inner learning rate | 0.4 | 0.05 | 0.05 | 0.05 |
| Meta learning rate | 0.001 | 0.001 | 0.001 | 0.001 |
| Meta-batch size | 4 | 4 | 4 | 4 |
| $K^{(tr)}$ meta-learning | 1 | 1 | 5 | 1 |
| $K^{(val)}$ meta-learning | 5 | 5 | 5 | 5 |
| $K^{(val)}$ evaluation | 15 | 15 | 5 | 15 |
| Meta-adaptation steps | 5 | 5 | 5 | 5 |
| Evaluation adaptation steps | 50 | 50 | 50 | 50 |

Table 6: LASIUM-ProtoNets hyperparameters summary

| Hyperparameter | Omniglot | CelebA | CelebA attributes | Mini-ImageNet |
|---|---|---|---|---|
| Number of classes | 5 | 5 | 2 | 5 |
| Input size | $28 \times 28 \times 1$ | $84 \times 84 \times 3$ | $84 \times 84 \times 3$ | $84 \times 84 \times 3$ |
| Meta learning rate | 0.001 | 0.001 | 0.001 | 0.001 |
| Meta-batch size | 4 | 4 | 4 | 4 |
| $K^{(tr)}$ meta-learning | 1 | 1 | 5 | 1 |
| $K^{(val)}$ meta-learning | 5 | 5 | 5 | 5 |
| $K^{(val)}$ evaluation | 15 | 15 | 5 | 15 |

We also report the ablation studies on different strategies for task construction in Table 7. We run all the algorithm for just 1000 iterations and compared between them. We also apply a small translation to Omniglot images.

Table 7: Accuracy of different proposed strategies on Omniglot. For the sake of comparison, we stop meta-learning after 1000 iterations. Results are reported on 1000 tasks with a 95% confidence interval.

| Sampling Strategy | Hyperparameters | GAN-MAML | VAE-MAML | GAN-Proto | VAE-Proto |
|---|---|---|---|---|---|
| LASIUM-N | $\sigma^2$=0.5 | **77.16$\pm$0.65** | 70.41 $\pm$ 0.71 | 62.16 $\pm$ 0.79 | 61.57 $\pm$ 0.80 |
| LASIUM-N | $\sigma^2$=1.0 | 71.10 $\pm$ 0.70 | 68.26 $\pm$ 0.71 | 60.95 $\pm$ 0.78 | 62.17 $\pm$ 0.80 |
| LASIUM-N | $\sigma^2$=2.0 | 63.18 $\pm$ 0.71 | 65.18 $\pm$ 0.71 | 59.81 $\pm$ 0.78 | **64.88$\pm$0.78** |
| LASIUM-RO | $\alpha$=0.2 | **77.62$\pm$0.64** | **75.02$\pm$0.66** | **62.24$\pm$0.79** | 62.17 $\pm$ 0.80 |
| LASIUM-RO | $\alpha$=0.4 | **75.79$\pm$0.65** | **71.31$\pm$0.70** | **64.19$\pm$0.76** | **62.20$\pm$0.80** |
| LASIUM-OC | $\alpha$=0.2 | 74.70 $\pm$ 0.68 | **74.98$\pm$0.67** | 61.79 $\pm$ 0.79 | 62.16 $\pm$ 0.78 |
| LASIUM-OC | $\alpha$=0.4 | 73.40 $\pm$ 0.68 | 68.79 $\pm$ 0.73 | **64.59$\pm$0.76** | **63.08$\pm$0.79** |

Besides, we perform a hyperparameter search on CelebA attributes benchmark. Table 8 demonstrates the results for our experiments. We see that searching for hyperparameters for CelebA is almost as easy as doing the same thing for Omniglot. LASIUM-N with $\sigma^2 = 0.25$ outperforms state-of-the-art in this benchmark. We also see a bad performance in the case of LASIUM-OC, which we expected as the number of classes in this benchmark's tasks is $N = 2$. Thus samples generated during meta-learning are limited to only instances on the line connecting two anchor latent vectors. It is not the case for LASIUM-N and LASIUM-RO since we can sample latent codes in the neighborhood or any direction from anchor points in the latent space.

---

[2]https://github.com/siavash-khodadadeh/MetaLearning-TF2.0

Table 8: Accuracy of different proposed strategies on CelebA attributes task for GAN with 2-way, 5-shot tasks with $K^{(val)} = 5$. The results are averaged over 1000 downstream tasks and $\pm$ indicates 95% confidence interval.

| Sampling Strategy | Hyperparameters | GAN-MAML | GAN-Proto |
|---|---|---|---|
| LASIUM-N | $\sigma^2$=0.1 | $71.83 \pm 1.08$ | $62.99 \pm 1.14$ |
| LASIUM-N | $\sigma^2$=0.25 | $75.07 \pm 1.08$ | $70.49 \pm 1.14$ |
| LASIUM-N | $\sigma^2$=0.5 | $71.41 \pm 1.13$ | $69.96 \pm 1.15$ |
| LASIUM-N | $\sigma^2$=1.0 | $60.37 \pm 1.01$ | $69.98 \pm 1.16$ |
| LASIUM-N | $\sigma^2$=2.0 | $50.00 \pm 0.00$ | $70.33 \pm 1.14$ |
| LASIUM-RO | $\alpha$=0.2 | $62.06 \pm 1.06$ | $62.73 \pm 1.18$ |
| LASIUM-RO | $\alpha$=0.4 | $67.57 \pm 1.11$ | $68.19 \pm 1.12$ |
| LASIUM-RO | $\alpha$=0.5 | $71.04 \pm 1.03$ | $68.94 \pm 1.12$ |
| LASIUM-OC | $\alpha$=0.25 | $59.69 \pm 1.11$ | $53.67 \pm 1.02$ |
| LASIUM-OC | $\alpha$=0.5 | $60.25 \pm 1.08$ | $56.05 \pm 1.08$ |

## 7.1 NEURAL NETWORK ARCHITECTURES

For Omniglot, our VAE model is constructed symmetrically. The encoder is composed of four convolutional blocks, with batch normalization and ReLU activation following each of them. A dense layer is connected to the end such that given an input image of shape $28 \times 28$, the encoder produces a latent vector of length 20. On the other side, the decoder starts from a dense layer whose output has length $7 \times 7 \times 64 = 3136$. It is then fed into four modules each of which consists of a transposed convolutional layer, batch normalization and the ReLU non-linearity. We use $3 \times 3$ kernels, 64 channels and a stride of 2 for all the convolutional and transposed convolutional layers. Hence, the generated image has the size of $28 \times 28$ that is identical to the input images. This VAE model is trained for 1000 epochs with a learning rate of 0.001.

Our GAN generator gets an input of size $l$ which is the dimensionality of the latent space and feeds it into a dense layer of size $7 \times 7 \times 128$. After applying a Leaky ReLU with $\alpha = 0.2$, we reshape the output of dense layer to 128 channels of shape $7 \times 7$. Then we feed it into two upsampling blocks, where each block has a transposed convolution with 128 channels, $4 \times 4$ kernels and $2 \times 2$ strides. Finally, we feed the outcome of the upsampling blocks into a convolution layer with 1 channel and a $7 \times 7$ kernel with sigmoid activaiton. The discriminator takes a $28 \times 28 \times 1$ input and feeds it into three $3 \times 3$ convolution layers with 64, 128 and 128 channels and $2 \times 2$ strides. We apply leaky ReLU activation after each convolution layer with $\alpha = 0.2$. Finally we apply a global 2D max pooling layer and feed it into a dense layer with 1 neuron to classify the output as real or fake. We use the same loss function for training as MSGAN (aka Miss-GAN) described in Mao et al. (2019).

For the CelebA GAN experiments, we use the pre-trained network architecture, progressive growing of GANs (ProGAN), described in Karras et al. (2018). For VAE, we use the same architecture as we described for Omniglot VAE with one more convolution block and more channels to handle the larger input size of $84 \times 84 \times 3$. The exact architecture is described in the supplemental material.

## 7.2 IMPACT OF GAN TRAINING ON LASIUM

Do we need a generative model that generates very high-quality images from data or can a premature trained GAN also work? We performed an ablation study to evaluate the impact of GAN training on LASIUM. First, we trained a generative network on Omniglot dataset with adversarial training for 500 epochs and saved the corresponding weights at every epoch. Then we trained LASIUM with various generative networks at different epochs. For the sake of comparison, we stopped LASIUM after 1000 iterations.

Figure 5 demonstrates the impact of GAN training on LASIUM. We visualize an image generated with the same exact latent code after different epochs. We can see that eventually, this latent code result in generating character "R" (after epoch 400 and 500). We see that GAN stabilizes after 400 epochs while LASIUM stabilizes sooner around epoch 200. Nevertheless, the impact of training GAN for at least 50 epochs is correlated with LASIUM performance.

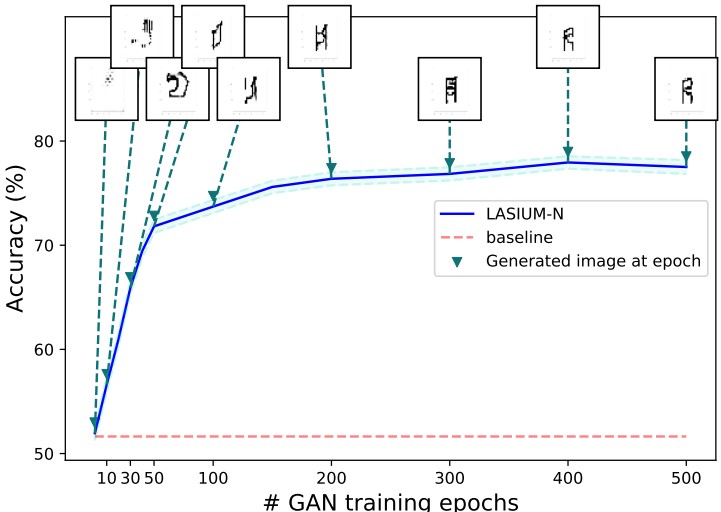

Figure 5: The impact of GAN training on LASIUM accuracy. The blue line shows test accuracy after 1000 iterations of LASIUM-N training with a 95% confidence interval on 1000 one-shot tasks with $K^{(val)} = 15$. The images generated by GAN are shown at epochs 0, 10, 30, 50, 100, 200, 300, 400, and 500. All of the images are generated from the same latent vector. The red line shows the training from scratch baseline.

### 7.3 ABLATION STUDY ON IMPACT OF $\epsilon$ ON LASIUM

In this section, we evaluate the accuracy of LASIUM with respect to the value of $\epsilon$. For the sake of comparison, we consider LASIUM-N with $\sigma^2 = 0.5$ and stop the training after 1000 iterations on Omniglot. The results are reported on the same 1000 one-shot tasks with $K^{(val)} = 15$ in Table 9. Furthermore, the last column shows the number of times resampling occurred since at least two of the initial sampled latent codes were in a distance smaller than $\epsilon$ from each other. We found that (within reasonable bounds) the choice of this hyperparameter has a small but not negligible impact on the performance of the algorithm.

### 7.4 TRAINING LASIUM ON FUNGI

We also tried LASIUM on Fungi dataset. We report LASIUM-N-GAN-MAML accuracy over 1000 downstream tasks generated randomly from test dataset following Meta-dataset evaluation protocol proposed by Triantafillou et al. (2020). For the choice of generative network, we used state-of-the-art StyleGAN-v2 by Karras et al. (2020), and we trained it on Fungi images. Figure 6 shows some of the examples generated by StyleGAN-v2. Table 10 shows the results on LASIUM and some other relevant algorithms.

Table 9: Accuracy of LASIUM-N with $\sigma^2 = 0.5$ on Omniglot dataset with respect to different values of $\epsilon$. $\pm$ indicates 95% confidence interval.

| $\epsilon$ | Accuracy (%) | # Resampling Task |
|---|---|---|
| 0.0 | $77.27 \pm 0.62$ | 0 |
| 0.1 | $77.34 \pm 0.62$ | 0 |
| 1 | $77.21 \pm 0.62$ | 0 |
| 10.0 | $77.54 \pm 0.62$ | 0 |
| 100.0 | $77.08 \pm 0.63$ | 0 |
| 125.0 | $79.51 \pm 0.61$ | 0 |
| 187.5 | $78.87 \pm 0.60$ | 395 |
| 218.75 | $77.95 \pm 0.62$ | 6012 |
| 234.375 | $77.15 \pm 0.63$ | 27432 |
| 242.1875 | $77.15 \pm 0.64$ | 61118 |
| 250.0 | $78.49 \pm 0.61$ | 155499 |
| 253.15625 | $78.48 \pm 0.62$ | 238714 |
| 256.3125 | $78.05 \pm 0.62$ | 378742 |
| 265.625 | $78.56 \pm 0.61$ | 1472860 |
| 281.25 | $77.73 \pm 0.63$ | 25936957 |
| 312.5 | – | All |
| 375 | – | All |
| 500.0 | – | All |
| 1000.0 | – | All |

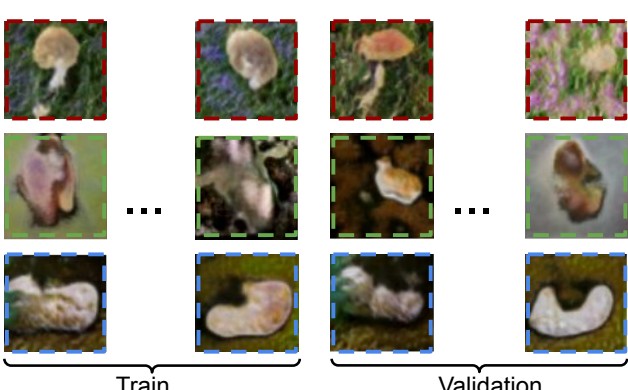

Figure 6: Meta-training tasks for Fungi constructed by LASIUM-N with $\sigma^2 = 0.25$

Table 10: Accuracy of 5-way 1-shot learning on the Fungi dataset (part of the proposed Meta-dataset by Triantafillou et al. (2020)). For each system we indicate the dataset on which the meta-training phase was performed. The results for supervised first-order MAML are from Triantafillou et al. (2020). LASIUM-N was run with $\sigma^2 = 0.25$ and used the StyleGAN-v2 trained on the unsupervised version of the Fungi dataset, as discussed in the text.

| Method | Dataset | Accuracy (%) |
|---|---|---|
| Training from scratch | - | $26.10 \pm 0.42$ |
| fo-UMTRA | Unsupervised Fungi | $28.27 \pm 0.46$ |
| LASIUM-N-GAN-fo-MAML | Unsupervised Fungi | $29.43 \pm 0.49$ |
| LASIUM-N-GAN-MAML | Unsupervised Fungi | $\mathbf{31.29 \pm 0.52}$ |
| fo-MAML | Supervised Imagenet | $32.10 \pm 1.10$ |
| fo-MAML | Supervised Meta-dataset | $33.54 \pm 1.11$ |

