# OpenReview forum: "Unsupervised Meta-Learning through Latent-Space Interpolation in Generative Models"
_ICLR.cc/2021/Conference — ICLR 2021 Poster_

### Official Review · AnonReviewer3 · 2020-10-21
**Review of "Unsupervised Meta-Learning through Latent-Space Interpolation in Generative Models"**

**Rating:** 6
**Confidence:** 4

**Review:**

This paper presents a novel method for generating few-shot learning tasks from unlabeled data. The basic idea is to use a GAN or VAE to generate in-class (close in latent space) and out-of-class (far away in latent space) pairs of image data by linear interpolation in the latent space of the chosen generative model. A new in-class sample is generated by interpolating between two out-of-class samples in latent space with the interpolation weight chosen such that the generated sample is close to one of the out-of-class samples. The method is evaluated on common few-shot learning datasets with good results.

**Pros:**
- The concept of generating few-shot learning tasks from an unlabeled dataset by performing interpolation in the latent space of a GAN or VAE is innovative and promising.
- The experiments demonstrate that the proposed method is as good as or superior to competitive unsupervised methods in basic scenarios.
- The paper is very well written, and the ideas are clearly presented.
- The diagrams are very helpful to understanding the basic concepts.

**Concerns:**
(1) The experiments are basic and do not explore potential limitations of the proposed method:
- All the experiments were performed with a low, fixed value of way equal to 5 and a fixed number of shots on datasets with homogeneous content. It would be interesting to see the results of using LASIUM on the more challenging meta-dataset [1] which uses tasks of variable way and shot on 10 diverse datasets. In a brief examination of the source code supplied in the supplementary material, it seems that some experimentation on meta-dataset may have been done. Any reason that those results were not presented?
- In particular, some of the datasets within meta-dataset (e.g. Fungi has a large number of classes of mushrooms that differ only subtly) require very fine-grained discrimination. It would be interesting to evaluate LASIUM in a higher way (> 25), fine-grained classification setting where classes are likely to be close in the latent space of the generator.

(2) In section 2, it says: “Our proposed approach, LASIUM, takes advantage of generative models trained on the specific domain to create the in-class and out-of-class pairs of meta-training data.” However, the goal of a few-shot learning system after meta-training is that the system should be able to make predictions on unseen input data, potentially from a different domain. As pointed out in (1) above, LASIUM has only been evaluated on a single domain. How would LASIUM handle a setting with very diverse tasks? Would more than one pre-trained GAN be needed to generate the examples? Or perhaps a GAN that has be trained on a large quantity of diverse data?

(3) LASIUM uses the latent space of a GAN or VAE to linearly interpolate between examples to generate few-shot learning training data. Were any ablation studies performed to ascertain that this is the best space for generating data? As a counter example, [3] uses linear interpolation in the raw data domain to augment training data in the supervised setting (i.e. they do not evaluate the method in the unsupervised setting) with particularly good results. The authors of [3] also consider doing the interpolation in an embedded feature space. Linear interpolation in alternative data spaces would make for a potentially interesting baseline or ablation study.

(4) Some experiment details are missing:
- The main text (including Algorithm 1) makes no mention of using labeled data. However, in Section 4.2 it says: “For instance, the 95.29% accuracy of LASIUM-RO-GAN-MAML was obtained with only 25 labels, while the supervised approaches used 25,000.” Please elaborate on the use of labels in the various methods. Are some labels used to establish “anchor vectors”? How do the results change if you use no labels (i.e. fully unsupervised as opposed to semi-supervised)? Was labeled data also used by the competitive unsupervised approaches (i.e. CACTUs, UMTRA)?
- No detail was given on how the transfer learning results were obtained (i.e. learning rates, number of iterations, regularization, augmentation, etc.). Specifically, the concern is that the transfer learning results seem to be quite low, while [2] shows that supervised transfer learning can outperform supervised few-shot learning methods. Also, the transfer learning results for miniImageNet seem to be missing.
- What values for $\alpha$ were used in the experiments? In table 6 in the supplementary material, there is a small ablation study for selecting a good value $\alpha$ for the Omniglot experiments. It would be good to see this done for a greater range of $\alpha$ and for datasets other than Omniglot. The key question is to know how difficult it is to set $\alpha$ for each setting.
- What values are used for the rejection sampling threshold $\epsilon$ for the various methods? How sensitive is this parameter?
- For the sake of reproducibility, details on how all the baseline experiments were carried out would be beneficial.

(5) Some details are missing from Algorithm 1 including: a) how the unlabeled dataset gets encoded to the latent space in the VAE case and what role the unlabeled dataset plays in the GAN case; and b) the required hyperparameters for the sampling strategy (i.e. $\alpha$, $\sigma$, $\epsilon$).

**References:**
[1] Triantafillou, Eleni, et al. "Meta-dataset: A dataset of datasets for learning to learn from few examples." arXiv preprint arXiv:1903.03096 (2019).
[2] Tian, Yonglong, et al. "Rethinking Few-Shot Image Classification: a Good Embedding Is All You Need?." arXiv preprint arXiv:2003.11539 (2020).
[3] Zhang, Hongyi, et al. "mixup: Beyond empirical risk minimization." arXiv preprint arXiv:1710.09412 (2017).

---

> ### Author Response · Authors · 2020-11-24
> **Author Response to AnonReviewer3**
>
> We thank the reviewer for the constructive feedback.
>
> Answers to comments in the concerns section:
>
> Comment 1: The experiments are basic and do not explore the potential limitations of the proposed method.
> Answer: We agree that a wide range of other comparison studies could be made.
> - We prioritized the benchmarks where we can do apples-to-apples comparisons with the other unsupervised meta-learning algorithms. We are comparing ourselves against (Omniglot, CelebA, Mini-ImageNet).
> - Any reference to meta-dataset comes from the generic meta-learning framework we developed. As the reviewer correctly points out, the Fungi dataset is much more fine-grained compared to, for example, Mini-ImageNet. We ran LASIUM-N on Fungi. The details are in section 6.4. We thank the reviewer for this suggestion that can better show the impact of our method.
> - "Higher N values": Based on studies on meta-learning literature (MAML, ANIL, etc), increasing N in meta-learning leads to a rapid decrease in performance. As unsupervised meta-learning is upper bounded by supervised meta-learning, we expect the same thing to happen for LASIUM.
>
> Comment 2: How would LASIUM handle a setting with very diverse tasks? Would more than one pre-trained GAN be needed to generate the examples? Or perhaps a GAN that has be trained on a large quantity of diverse data?
>
> Answer: This is a very interesting point, and it is an open question for meta-learning in general. Meta-learning algorithms, even in a supervised setting, are not robust to domain shifts. For example, a model that was meta-trained on CelebA performs very badly on tasks drawn from Mini-Imagenet and vice versa. We believe that this is related to the fact that the granularity of the classes is very different: what is one class (person) in MS-COCO is split into >1000 classes in CelebA etc. So, the current state-of-the-art in meta-learning requires the meta-training tasks to be roughly similar to the target task.
>
> The way in which this translates to LASIUM as an example of unsupervised meta-learning that generates the meta-training tasks is that we need a generative model trained on the domain itself, or on a closely related domain.
>
> Comment 3: Relationship to "mixup".
>
> We updated the related work to point out the similarity between LASIUM-OC and mixup - both of them generate new training examples through interpolation. However, the differences are many: mixup is not used for few-shot learning, it is not a meta-learning model, it is supervised, operates in pixel space (vs latent space) and the interpolation is not used to generate new in-class examples but examples that are interpolated in the label-space as well. Due to this list of differences, it is not easy to perform a comparison experiment in the limited time period of rebuttal, but we are thankful to the reviewer for introducing this paper and will certainly explore it in our future research.
>
> Comment 4.1:
> Supervised meta-learning approaches like the original MAML, use human-labeled tasks in the meta-learning process. The output of this "learning-to-learn" process is a few-shot learner. If the setting is 5-way 1-shot, it would require 5 labeled samples, one of each class.
> Unsupervised meta-learning approaches like CACTUs, UMTRA and LASIUM do not use any label in the meta-learning phase. The output is still a few-shot learner, which needs a small number of labeled samples.
>
> Comment 4.2
> We added the transfer learning results to the Mini-Imagenet dataset and included the table in the main paper. As expected transfer learning outperforms all unsupervised meta-learning methods. In the case of Mini-ImageNet, it outperforms supervised MAML and protonets.
> To put things in perspective, VGG19 is a network with 19 layers which takes images of shape 224 x 224 x 3 and had been trained on more than 1M images on ImageNet. All the meta-learning techniques in the comparison use a much smaller network of just four convolution layers on an input of 84 x 84 x 3.
>
> Comment 4.3: Thank you for the suggestion. We performed an ablation study on CelebA and the results are added in supplemental material now in section 6. Indeed, by performing a binary search on the value of $\sigma^2$ we found a value of $\sigma^2$ for LASIUM-N that consistently outperforms CACTUs.
>
> Comment 4.4: As suggested, we run a series of experiments for various values of  $\epsilon$.  The results of these experiments are in Subsection 6.3 and Table 9 in the supplemental material. We found that (within reasonable bounds) the choice of this hyperparameter has a small but not negligible impact on the performance of the algorithm.
>
> Comment 4.5: We will release our code (including both LASIUM and baselines) and the pre-trained models on Github.
>
> Comment 5: We updated Algorithm 1 in the paper to specifically identify the fact that the hyperparameters (i.e. $\alpha$, $\sigma$, $\epsilon$) and the pre-trained generator is an input to the algorithm.

---

### Official Review · AnonReviewer4 · 2020-10-27
**OK, but miniImageNet results are missing.**

**Rating:** 6
**Confidence:** 5

**Review:**

#### Summary
- This paper proposes LASIUM, a framework for constructing tasks for unsupervised meta-learning for image classification via the use of a generative model on the unlabeled training data. Similar to Hsu et al. (2019), distance in the latent space of an unsupervised learner is taken to correspond to class-level information. Uniquely, LASIUM leverages generative modeling by proposing simple interpolation-based schemes to populate classes with latents, which are decoded by the generative model to produce image samples for the meta-learning stage. Three specific variants of LASIUM are proposed. The results indicate that the approach is competitive with respect to prior methods for the Omniglot and CelebA datasets, but curiously, results on miniImageNet are not presented.

#### Strengths
- The idea of using generative modeling and interpolation for unsupervised meta-learning for image classification is significantly different from prior works, which use data augmentation (Antoniou and Storkey 2019, Khodadadeh et al. 2019) and latent-space clustering (Hsu et al., 2019).

- Reasonable variations of the core idea of latent-space interpolation and generation are explored and assessed. This includes a VAE-specific variant which additionally leverages the encoding capability of a VAE.

- The experiments involve a reasonable variety of generative models, and the protocol seems to closely follow that of prior work.

#### Weaknesses
- Critically, experiments based on the miniImageNet dataset are completely missing. This is a benchmark that is present in all three prior works in unsupervised meta-learning for image classification ((Antoniou and Storkey 2019, Hsu et al. 2019, Khodadadeh et al. 2019), as well as the vast majority of meta-learning works. The diversity present in miniImageNet makes it a benchmark that assesses significantly different model capabilities than Omniglot or CelebA, and hence makes it an indispensable part of a comprehensive empirical evaluation.

- Unfortunately, there is a fair amount of overclaiming. "The proposed family of algorithms generate pairs
of in-class and out-of-class samples from the latent space in a principled way" -- the authors do not go beyond high-level intuition in justifying the latent interpolation schemes for task generation. "Beyond the increase in performance, we must note that the competing approaches use more domain specific knowledge (in case of UMTRA augmentations, in case of CACTUs, learned clustering)." -- how does CACTUs use more domain-specific knowledge than LASIUM? Both methods rely on an unsupervised pre-training stage to "organize" the unlabeled dataset into a latent space.

- The use of generative models introduces a significant potential weakness: the performance of the downstream meta-learning is now additionally dependent on sample fidelity. CACTUs sidesteps this issue by only using samples from the original dataset, and AAL and UMTRA rely on carefully defined image transformations that do not compromise sample quality. This is touched upon in the discussion and described as a "limitation", but this key issue is not explored further.

- There are significant clarity issues with the discussion section. "A conceptual difference between clustering and LASIUM" -- the difference is really between CACTUs and LASIUM; clustering doesn't entail encoding by itself. "The former is shared with clustering based unsupervised learning" -- what is clustering based unsupervised learning? Do you mean CACTUs? Also, do you mean "latter" instead of "former"? "Interestingly, training of these feature extractors is based on augmentations like random crop and random flips (Caron et al. (2018)) or auto encoders (Berthelot et al. (2019))" -- what precisely is interesting about this? Do you mean to make an explicit connection between CACTUs, UMTRA/AAL, and LASIUM here?

#### Recommendation
- I currently recommend rejection (4). The weaknesses outlined above outweigh the strengths.

#### Questions
- Why are results on miniImageNet not included?

- Under what conditions can we expect LASIUM to be a favorable choice over CACTUs or UMTRA/AAL, if any?

- "Note that meta-learning algorithms are not sensitive to a small fraction of bad training examples in the meta-training phase (e.g. see Hsu et al. (2019); Khodadadehet al. (2019))." -- How do these works demonstrate this?

#### Minor suggestions
- There are minor typos, e.g. "proGAN" instead of "ProGAN".

------------------------------------ Post-rebuttal comments ------------------------------------

With the authors' answers, the inclusion of miniImageNet results in the main text, and revisions to the writing, my concerns have largely been addressed and I have increased my rating from a 4 to a 6.

I have a few further suggestions to make: first, make it clear in Table 4 via captioning and/or markings that UMTRA and LASIUM results for miniImageNet depend on the use of full unlabeled ImageNet data to train the AutoAugment and BigBiGAN models, respectively, while the CACTUs results do not. This results in an unfair comparison, and naturally raises the question of how LASIUM would do if we used a generative model trained only on the meta-training split of miniImageNet, which would be more in line with the protocol used for Omniglot and CelebA. It seems fair to assume that it would do worse than the current LASIUM results in Table 4, and probably the CACTUs results as well. This relates back to my point in the original review about sample fidelity: miniImageNet has the most diversity among the datasets considered, and it is hard for generative models to capture this diversity given a relatively small dataset. It would be good for this to be conveyed to the reader.

The usage of the extraneous ImageNet data for UMTRA and LASIUM does not conform to the definition of unsupervised meta-learning proposed in Hsu et al. (2019) and adopted in this work. Some discussion of the problem assumption may be warranted: in what practical circumstances would extraneous, relevant, unlabeled data be available? And when it is, why would one not use all of the data (e.g. the entirety of unlabeled ImageNet) to do unsupervised meta-learning and/or unsupervised representation learning, like in Table 12 of Hsu et al. (2019)?

Overall, despite the unfair miniImageNet protocol, I still advocate for weak acceptance as the method does show competitive results for domains that are more suited to generative modeling. For the sake of clarity for readers, though, I strongly encourage the authors to implement my additional recommendations.

---

> ### Author Response · Authors · 2020-11-24
> **Author Response to AnonReviewer4**
>
> We thank the reviewer for valuable feedback.
>
> Response to questions:
> 1) The miniImageNet results were in the supplemental material, Section 7. We apologize for the confusion. We moved these results to the main paper (see section 4.4 and Table 4).
>
> 2) In our experiments, LASIUM outperformed CACTUs in the majority of cases and UMTRA in all cases. Another consideration is ease of deployment. Let us assume that a pre-trained generative model is available for the domain itself or for a closely related domain. Then creating meta-learning data through LASIUM-N only requires choosing the class size parameter $\sigma$. This requires less engineering than choosing the right augmentations or clustering method.
>
> 3) With regards to meta-learning algorithms not being sensitive to a small fraction of bad training examples, we are relying on the following parts of the referred papers:
>
> Hsu et al. (2019) (CACTUs): Figure 2 (a) top right and bottom right. The caption mentions that some clusters are uninterpretable (top right) or are based on image artifacts (bottom right). These clusters clearly are not going to help with downstream tasks. However, one can assume that the number of such generated tasks are going to be small.
>
> Khodadadeh et al (2019) (UMTRA): In section 3.2 formula (1), the authors calculate the probability of sampling instances from the same class as different classes during training. They show that this probability is small but not zero. This means that a fraction of generated tasks could be tasks that are not useful, however, the meta-learning algorithm is still robust to these and outperforms other methods.
>
> Answers to further questions listed in the weaknesses section:
>
> *) "a significant potential weakness: the performance of the downstream meta-learning is now additionally dependent on sample fidelity"
>
> We agree that the meta-learning performance depends on the quality of the generative model. This is not different from any other approach where the performance depends on the quality of the augmentation, or clustering etc.
> To quantify this we performed a new series of experiments, investigating the evolution of the performance with respect to the number of epochs the GAN was trained. The results are in section 6.2 in the supplemental material. We find that while a correlation exists in the early phases of training, LASIUM works well with a partially trained generative model. For instance, Figure 5 shows that at epoch 100, LASIUM already achieves most of the performance benefits, while the visual quality of the samples is still very bad.
>
> *) Comment: There are significant clarity issues with the discussion section.
>
> We have removed the speculative statements from the discussion section.
>
> *) Comment about: "Beyond the increase in performance, we must note that the competing approaches use more domain specific knowledge..."
>
> Clearly, in order to achieve equivalent performance equivalent knowledge is needed. The difference is how much of this knowledge must be entered by the practitioner during the meta-training phase.
> Our rationale for this claim was that knowledge in the form used by LASIUM (in the form of a pretrained generative model) appears more readily accessible than in the form used by UMTRA and AAL (augmentations) or CACTUs (clusters). But maybe this is simply our perspective due to the availability of high quality pre-trained GANs.
> We decided to remove this claim, as it might be more suitable for an informal discussion, eg. on a blog post etc.

---

> ### Author Response · Authors · 2021-03-12
> **Author Response to AnonReviewer4 - Post Rebuttal**
>
> Thank you for your suggestions and for increasing the review score. We believe that during the rebuttal, the quality of our paper improved a lot as a result of the constructive feedback we received from all the reviewers. We will definitely apply the changes you suggested in the camera-ready version.

---

### Official Review · AnonReviewer2 · 2020-10-28

**Rating:** 6
**Confidence:** 4

**Review:**

This work proposes LASIUM (LAtent-Space Interpolation Unsupervised Meta-learning), a novel method of generating meta-tasks in an unsupervised way. It leverages recent advances in generative models. Specifically, it uses the fact that datapoints that are close in latent space are more likely to be from the same class. The method interpolates between samples in latent space, choosing interpolation ratios.

I find this work compelling: it is built upon simple and intuitive ideas and leverages a rapidly growing research area. Its ability to use any pre-trained generative model to perform unsupervised meta-learning is promising.

The paper proposes three different variants of LASIUM, each of them compelling ideas: LASIUM -N, -RO, -OC. However, the paper does not comment on, which is best or what tradeoffs exist between the three methods. RO seems to win on Omniglot, but the paper only shows N for the hardest task (CelebA).

It would have been helpful to see how the generative model affects unsupervised meta-learning performance. For example, is being good at image generation always better for meta-learning? We cannot know from the experiments presented here because it only tries one model for each setting.

Minor comments
- the underlined phrases scattered throughout the paper helped comprehension. Nice touch.
- (very minor) page 4: inconsistent indentation for paragraph "Generating in-class...". Other bolded paragraphs are not indented.
- Table 3: Missing digit (probably 0) on MLP with dropout.

---

> ### Author Response · Authors · 2020-11-24
> **Author Response to AnonReviewer2**
>
> We thank the reviewer for constructive feedback. Please note that LASIUM-N is the simplest approach which just samples from a small neighborhood. LASIUM-RO, uses a meaningful direction in generating samples while reducing finding a proper $\sigma^2$ to finding a coefficient between 0 and 0.5. LASIUM-OC, uses meaningful direction with respect to other samples in the task to generate harder validation sets. We added ablation studies for CelebA attribute classification benchmark in Table 8 in the supplemental material. Thanks to the reviewer’s comment on this, we were able to systematically find a better hyperparameter that can outperform the CACTUs method proposed by Hsu et al. (2019) on this benchmark. We also added a detailed explanation about this hyperparameter search experiment in section 6 of supplemental material.
>
> To answer the question of the impact of the quality of the generative model on the classification performance, in the new Section 6.2 we compared the performance of LASIUM using a GAN in different stages of training. As Figure 5 shows, the impact of the GAN training increases the accuracy very quickly in the early phases. Interestingly, however, LASIUM can still perform very well with a GAN which is not yet fully trained. For instance, at epoch 100, the GAN generates a figure barely recognizable as a letter, yet most of the LASIUM benefits had been already realized.
>
> Finally, we addressed the minor comments and thank the reviewer for their feedback and interesting proposed ablation studies.

---

### Official Review · AnonReviewer1 · 2020-11-05
**Interesting work for unsupervised meta-learning**

**Rating:** 7
**Confidence:** 4

**Review:**

This paper considers the problem of unsupervised meta-learning, where the goal is to generate tasks for meta-training without supervision. Whereas previous work generated training and test sets from the unlabeled set for meta-training via augmentations (UMTRA) or unsupervised clustering of embeddings (CACTUs), this paper considers doing this using interpolation of the latent space representations produced by generative models. Specifically, the idea is to first train a generative model on the unlabeled set and then produce training and test sets for meta-training by decoding the interpolation of latent space representations of multiple examples from the original unlabeled set. The authors discuss 3 specific ways to produce examples for the train and test sets for meta-training in this way. They first select an anchor example from the unlabeled set that will be representative of one class in the dataset. Then, the 3 methods involve:
1. Adding noise: adding noise to the latent space representation of the anchor example to produce examples that make up the training and test set for a single class.
2. Random out-of-class sample: selecting another example from the unlabeled set and finding new examples for the class by interpolating between the anchor's and this example's representations.
3. With Other Classes' samples: same as (2) but instead of picking another random example, the example considered is another anchor example that was used to represent a different class.

The authors evaluate their method by considering 3 few-shot learning benchmarks: (1) Omniglot; (2) CelebA few-shot identity recognition; and (3) CelebA attribute prediction. On these 3 benchmarks, they show that their method performs favorably compared to UMTRA and CACTUs.

Pros
* This paper proposes a simple yet very interesting idea for performing unsupervised meta-learning that is unique compared to previous work.
* The big benefit of this method compared to previous work is that it seems to require less tweaking per dataset. Whereas previous methods required tuning per dataset (for example, in UMTRA, selecting which augmentations to use for a specific dataset), this method requires training a generative model on the unlabeled set and using the learned latent space interpolation (where the generative model can directly learn properties of the specific dataset that can be used during interpolation). However, there are still some choices to make in terms of hyperparameters for latent space interpolation).

Cons
* I have minor concerns about the Mini-ImageNet experiments. Firstly, why are Mini-ImageNet experiments not discussed in the main paper but in the supplementary material? The difficulties of using Mini-ImageNet are mentioned, in that it is difficult to train a generative model on this more complex dataset using the limited examples in Mini-ImageNet. Thus, I believe it is a good idea to use whole ImageNet dataset as the unlabeled set, as the authors did, and the results for the method seem favorable compared to previous work. So, I think it's useful to include these results to show how this method extends to more complex images? I think a note just needs to be added that the unlabeled set this method uses is much larger than the ones used in previous work for the Mini-ImageNet comparison but I don't view this as a big negative because the data required for training is still unlabeled.
* Details of the CelebA few-shot identity recognition benchmark seem to be lacking? I don't see this benchmark mentioned in previous work so I was curious how the metrics for other methods (such as CACTUs) were generated given that this benchmark wasn't discussed in those papers? I think more details about this benchmark would be useful in general. Additionally, some citations to previous results on this benchmark would also be helpful.

---

> ### Author Response · Authors · 2020-11-24
> **Author Response to AnonReviewer1**
>
> We thank the reviewer for constructive feedback. As you suggested, we included the Mini-ImageNet results into the main paper.
>
> We believe that using identity recognition is a natural application field for few-shot learning, and a large number of labeled identities in CelebA makes it a good test for it. For the comparison, we used the available public code for CACTUs and UMTRA and performed all the test experiments on the same exact 1000 tasks to make sure that our comparisons are fair. We will release the code for LASIUM with instructions containing all benchmark details upon publication.

---

### Author Response · Authors · 2020-11-24
**Summary of changes**

We sincerely thank the reviewers for their constructive feedback and suggestions. Here is a brief list of changes we made in the manuscript based on these suggestions:

- As suggested by AnonReviewer1 and AnonReviewer4 we now include the Mini-Imagenet results in the main paper (section 4.4). This previously was in the supplemental material.
- We added a new related work [1] to the manuscript.
- We revised the discussion section and removed the speculative statements for better comprehensibility.
- We added hyperparameter search on CelebA attributes benchmark. The details are now in table 8 and supplemental material section 6.
- We added a section to supplemental material and a figure (Figure 5)  that visualizes the relation between gan generated samples and the accuracy of LASIUM that was asked about by reviewers (Please look at section 6.2. and figure 5 in the supplemental material).
- We show the effect of $\epsilon$ on LASIUM in section 6.3 of supplemental material. Table 9 shows the accuracies and corresponding $\epsilon$ value.
- Finally, following the reviewers’ suggestion, we demonstrate the results of LASIUM on the Fungi dataset that is one of the datasets in meta-dataset [2]. (see section 6.4 for more details).

[1] Hongyi Zhang, et al. "mixup: Beyond empirical risk minimization." arXiv preprint arXiv:1710.09412 (2017).

[2] Eleni Triantafillou, et al. Meta-Dataset:  A Dataset of Datasets for Learning to Learn from Few Examples.  In Int’l Conf. on Learning Representations (ICLR), 2020.

---

### Decision · Program_Chairs · 2021-01-07
**Final Decision**

**Decision:**

Accept (Poster)

**Comment:**

This paper addresses a method for generating meta-tasks via latent space interpolation using a generative model, trained on the unlabeled dataset, to solve the unsupervised meta-learning. The method seems to be sound, but it lacks MiniImage-Net experiments, which is the main concern raised by most of reviewers. During the author responses, new empirical results on MiniImage-Net were added. During the discussion period with reviewers, I communicated with the reviewer with most negative comments. He/she was not fully satisfied, claiming that the protocol used for obtaining new MinImage-Net results was slightly unfair. It is suspected that the authors used a generative model trained on the ImageNet training set (of which miniImageNet is a subset) for LASIUM and  did not present results from only using miniImageNet meta-training data because it was not competitive with prior work. It should be clarified in the final paper. However, we arrived at the consensus that the paper is worth being presented.